# Combination Antioxidant/NSAID Therapies and Oral/Topical Ocular Delivery Modes for Prevention of Oxygen-Induced Retinopathy in a Rat Model

**DOI:** 10.3390/nu12071980

**Published:** 2020-07-03

**Authors:** Kay D. Beharry, Charles L. Cai, Faisal Siddiqui, Christina D’Agrosa, Anano Zangaladze, Ghassan Mustafa, Areej Qadri, Thomas J. Duggan, Jacob V. Aranda

**Affiliations:** 1Division of Neonatal-Perinatal Medicine, Department of Pediatrics, State University of New York Downstate Medical Center, Brooklyn, NY 11203, USA; Charles.cai@downstate.edu (C.L.C.); Faisal.siddiqui@downstate.edu (F.S.); anano.zangaladze@downstate.edu (A.Z.); ghassan.mustafa@downstate.edu (G.M.); Areej.qadri@downstate.edu (A.Q.); Thomas.duggan@downstate.edu (T.J.D.); Jacob.aranda@downstate.edu (J.V.A.); 2Department of Ophthalmology, State University of New York Downstate Medical Center, Brooklyn, NY 11203, USA; 3SUNY Eye Institute, State University of New York Downstate Medical Center, Brooklyn, NY 11203, USA

**Keywords:** coenzyme Q10, insulin-like Growth Factor-I, glutathione nanoparticles, *n*-3 polyunsaturated fatty acids, ketorolac, neonatal intermittent hypoxia, oxygen-induced retinopathy, vascular endothelial growth factor

## Abstract

Given the complexity of oxygen-induced retinopathy (OIR), we tested the hypothesis that combination therapies and modes of administration would synergistically optimize efficacy for prevention of OIR. Newborn rats were exposed to neonatal intermittent hypoxia (IH) from the first day of life (P0) until P14 during which they received: (1) oral glutathione nanoparticles (nGSH) with topical ocular phosphate buffered saline (PBS); (2) nGSH with topical ocular Acuvail (ACV); (3) oral coenzyme Q10 (CoQ10) + ACV; (4) oral omega 3 polyunsaturated fatty acids (*n*-3 PUFAs) + ACV; (5) CoQ10 + *n*-3 PUFAs + PBS; or (6) CoQ10 + *n*-3 PUFAs + ACV. Treated groups raised in room air (RA) served as controls. At P14, pups were placed in RA with no treatment until P21. Retinal vascular pathology, ocular angiogenesis biomarkers, histopathology, and morphometry were determined. All combination treatments in IH resulted in the most beneficial retinal outcomes consistent with suppression of angiogenesis growth factors during reoxygenation/reperfusion and no significant adverse effects on somatic growth. nGSH + PBS also reversed IH-induced retinopathy, but had negative effects on growth. Simultaneously targeting oxidants, inflammation, and poor growth mitigates the damaging effects of neonatal IH on the developing retina. Therapeutic synergy with combination delivery methods enhance individual attributes and simultaneously target multiple pathways involved in complex diseases such as OIR.

## 1. Introduction

Retinopathy of prematurity (ROP) is a potentially blinding disease that afflicts predominantly extremely low gestational age neonates (ELGANs) who are ≤28 weeks gestation [1,2,3,4]. The etiology of ROP is complex and involves the use of excessive and high levels of supplemental oxygen, oxidative distress, inflammation, poor nutrition, dysregulated growth factors, and neonatal intermittent hypoxia (IH). In this regard, no single therapy has been proven to be successful without significant short- or long-term adverse effects. We have previously shown that combination treatment with systemic caffeine and topical ocular ketorolac resulted in prevention of severe IH-induced retinopathy in a rat model [5,6,7]. We have also shown that supplementation with either the antioxidant, coenzyme Q10 (CoQ10) or omega 3 polyunsaturated fatty acids (*n*-3 PUFAs), targeting oxidants and/or growth deficits, were individually effective for ameliorating specific characteristics consistent with ROP [8]. However, these individual supplements did not completely prevent severe oxygen-induced retinopathy (OIR) and many unwanted retinal pathologies persisted. Based on their individual therapeutic benefits, the current study was conducted to determine whether combined treatments and mode of delivery, simultaneously targeting the key underlying pathways, i.e., oxidative distress, inflammation, and poor growth would optimize their individual mechanistic attributes, reduce barrier limitations, and therapeutically synergize to prevent severe IH-induced OIR.

The first target was oxidative distress/oxidative aggression and lipid peroxidation to which ELGANs are highly susceptible due to supplemental oxygen, mechanical ventilation, immature antioxidant systems, reduced placental transfer of antioxidants due to preterm birth, and immature plasma transferrin. Oxidative distress involves the production of reactive oxygen species (ROS) which remove or accept electrons from other molecules, thus reducing or oxidizing them. ROS include superoxide anion, the main precursor to many ROS, hydrogen peroxide (H_2_O_2_), nitric oxide (NO), peroxynitrite, and the hydroxyl radical. Superoxide anion is scavenged by superoxide dismutase (SOD) to form H_2_O_2_ and O_2_. Through the actions of catalase and glutathione peroxidase, H_2_O_2_ is then transformed to water. In ELGANs, supraphysiological levels of oxygen generate high amounts of ROS due to immature antioxidant systems and inefficient scavenging capacities. High levels of H_2_O_2_ can generate the hydroxl radical, which is highly reactive and harmful to nucleic acids, proteins, and lipids, in the presence of iron via the Fenton-Haber-Weiss reaction [9,10]. This is particularly noteworthy in preterm infants with low plasma transferrin and a reduced ability to bind excess iron [11,12,13,14]. CoQ10 is a powerful antioxidant that directly scavenges oxygen free radicals. In the mitochondria, CoQ10 is involved in oxidative phosphorylation (OXPHOS) and energy production [15,16]. CoQ10 is found in tissues with high energy demands including the eyes [17,18]. It has bioenergetic properties, is involved in energy production, and is involved in the prevention of membrane phospholipid peroxidation and free radical oxidation [19]. Numerous studies have shown the benefits of CoQ10 including reversing mitochondrial dysfunction and increasing cellular metabolism [20,21,22,23]. Glutathione (GSH) plays a key role as an antioxidant as it rapidly detoxifies H_2_O_2_, thus preventing its reaction with iron and the formation of the hydroxyl radical [24]. GSH is also a key regulator of lipid peroxides; its inactivation often results in accumulation of lipid peroxides and death [25,26]. However, antioxidant supplements alone may be inadequate to protect against oxidative distress as many human trials have shown [27,28,29]. 

Our second target in these experiments is poor growth. Parenteral supplementation of preterm infants with intravenous lipids provide a source of energy and essential fatty acids necessary for postnatal growth and brain development [30,31]. In addition to its growth-promoting benefits, *n*-3 PUFAs are anti-inflammatory [32]. *n*-3 PUFAs inhibit cyclooxygenase (COX) and lipoxygenase (LOX), which decreases proinflammatory prostaglandins and leukotrienes. Compared to other tissues, the retina has the highest concentration of *n*-3 PUFAs (20%), which is necessary for normal retinal architecture and function [33]. However, a systematic review and meta-analysis of clinical trials showed no benefits with *n*-3 PUFA supplementation for most neonatal diseases [34], and scientific data regarding the benefits of *n*-3 PUFA supplementation alone for ROP are contradictory [35,36,37,38,39]. Further, *n*-3 PUFA supplementation has been shown to enhance ROS production and cell damage [40,41]. This is likely due to the vulnerability of *n*-3 PUFAs to oxidation. 

Our final target was inflammation. Lipid peroxides are intermediates in the formation of prostaglandins and are important in inflammation and disease via the actions of COX, which peroxidizes linoleic acid [42] and cytochrome p450s (CYPs), which synthesizes epoxyeicosatrienoic acids (EETs) and LOX, which synthesizes lipid hydroperoxides and oxidation of arachidonic acid [43]. ROS play an important role in the release of AA from membrane phospholipids. In oxidative stress, free AA accumulates and undergoes uncontrolled oxidative metabolism by both enzymatic and nonenzymatic processes. This uncontrolled metabolism, referred to as the ‘‘arachidonic acid cascade’’, includes the formation of prostaglandins, leukotrienes, thromboxanes, isoprostanes, and nonenzymatic lipid peroxidation products [44]. The arachidonic acid cascade amplifies the overall production of ROS and subsequent oxidative damage to lipids, proteins, and nucleic acids [45]. In the eye, action of prostanoids cause vasodilatation, breakdown of the blood–ocular barrier, and subsequent hemorrhage. Increased production of thromboxane A_2_ (TxA_2_) via COX-2 can lead to time- and concentration-dependent retinal damage [46,47,48], particularly in the newborn retina [49]. Prostaglandins, particularly PGE_2_, can induce pathological retinopathy through binding to its EP3 receptor and production of vascular endothelial growth factor (VEGF) [50,51,52]. Topical administration of non-steroidal anti-inflammatory drugs (NSAIDs) is the preferred route of administration to prevent inflammation as it provides higher ocular drug concentrations, avoiding the systemic side effects of oral administration [53]. We [7] and others [54] have shown that topical ocular ketorolac is quickly absorbed by the ocular tissues when administered in single or in multiple doses in neonatal rats and was beneficial for preventing severe OIR. In preterm infants, topical ocular ketorolac was also shown to reduce the incidence of severe ROP [55]. These previous reports demonstrate the importance of oxidants, lipid mediators, and pro-inflammatory COX metabolites as potent modulators of proliferative retinopathies. Here we introduce a new treatment paradigm consisting of mechanistically distinct antioxidant, growth promoting, anti-inflammatory, and anti-angiogenesis modalities to test the hypothesis that their interactions would provide enhanced, synergistic benefits for prevention of severe OIR. 

## 2. Materials and Methods 

### 2.1. Animals

All experiments were approved by the State University of New York, Downstate Medical Center Institutional Animal Care and Use Committee, Brooklyn, NY (Protocol #19-10559). Animals were cared for according to the guidelines of the United States Department of Agriculture and the Guide for the Care and Use of Laboratory Animals. Certified infection-free, timed-pregnant Sprague Dawley rats were purchased from Charles River Laboratories (Wilmington, MA USA) at 18 days gestation. The animals were housed in an animal facility with a 12-hour-day/12-hour-night cycle and provided standard laboratory diet and water ad libitum until delivery of their pups. All procedures were performed in accordance with the Association for Research in Vision and Ophthalmology statement on the Use of Animals in Ophthalmic and Vision Research.

### 2.2. Experimental Design

Within 2–4 hours of birth, newborn rat pups delivering on the same day were pooled and randomly assigned to expanded litters of 18 pups/litter (9 males and 9 females), as previously described [5,6,8]. Gender was identified by the anogenital distance. The expanded litter size was used to simulate relative postnatal malnutrition of ELGANs who experience frequent arterial oxygen desaturations, intermittent hypoxia/hypoxemia episodes, or apnea of prematurity; and who are at increased risk for development of severe ROP. Animals were exposed to IH from P0 to P14, then allowed to recover in room air (RA) until P21 (reoxygenation/reperfusion). RA littermates were raised in atmospheric oxygen from P0 to P21 and served as controls. At P0, RA- and IH-exposed pups were randomized to receive: (1) oral nano glutathione sublingual drops (nGSH, 200 mg/mL, optimized for instant absorption) diluted to 24 µg in 50 µL with extra virgin olive oil from P0 to P14 and topical ocular sterile normal phosphate buffered saline (PBS, 5 µL) from P5 to P14. The dose of nGSH was based on the manufacturer’s recommended dose for a 70 Kg adult (Nanoceutical Solutions, San Antonio, TX, USA); (2) oral nGSH and topical ocular Acuvail (ACV, ketorolac ophthalmic solution, 4%, 5 µL) from P5 to P14. The dose of ACV was based on the manufacturer’s recommendation and adjusted for the neonatal rat; (3) 50 µL fish oil containing 35 mg total *n*-3 PUFAs (22 mg eicosapentaenoic acid, EPA and 13 mg docosahexaenoic acid, DHA) from P0-P14 and topical ocular ACV (5 µL) from P5 to P14; (4) CoQ10 (0.35 mg in 50 µL extra virgin olive oil) purchased from Sigma Aldrich (St. Louis, MO, USA) from P0-P14 and topical ACV (5 µL; ACV) from P5 to P14; (5) oral *n*-3 PUFAs + CoQ10 and topical ocular PBS; or (6) *n*-3 PUFAs + CoQ10 and topical ocular ACV. The doses of *n*-3 PUFAs and CoQ10 were based on results of our previous findings [8]. Total body weight (grams) and linear growth (crown to rump length, cm) were recorded at P0, P7, P14, and P21 to determine differences in weight accretion (calculated as weight or length at the end of the experiment (P21) minus weight or length at birth (P0) divided by the weight or length at P0 × 100. Percentage change in body weight was used to standardize differences in birth weight. Methods and daily doses of oral nGSH, CoQ10, and *n*-3 PUFAs were conducted as previously described [8]. Treatment occurred only from P0-P14 during IH exposure, and not during the reoxygenation/reperfusion period from P14 to P21. The caecal period in rats represents the period from conception to eye opening. This usually occurs at P14 and coincides with maturation of the retinal neural circuitry. The percentage of rats that opened their eyes at P14 was also recorded. At P21, animals were euthanized for collection of blood and eyes to determine systemic and ocular outcomes.

### 2.3. Neonatal Intermittent Hypoxia (IH) Profile

Animals randomized to IH were placed with the dams in specialized oxygen chambers (BioSpherix, NY, USA) attached to an oxycycler. These chambers were optimized for gas efficiency and provided adequate ventilation for the animals in a controlled atmosphere with minimal gas usage. Oxygen content inside the chamber was continuously monitored and recorded on a Dell Computer. Carbon dioxide in the chamber was monitored and removed from the atmosphere by placing soda lime within the chamber. The IH profile consisted of an initial exposure of hyperoxia (50% O_2_) for 30 min followed by three brief, 1-minute, clustered hypoxic events (12% O_2_), with a 10-minute re-oxygenation in 50% O_2_ between each hypoxic event. Recovery between each IH episode occurred in 50% O_2_ following each clustered IH event for 2.5 hours for a total of 8 clustering IH episodes per day for 14 days, as previously described [5,6,8,56,57,58]. This model is extensively used in our laboratory and simulates brief arterial oxygen desaturations experienced by ELGANs who are at risk of severe ROP. Oxygen saturation was confirmed on a sentinel unanesthetized rat pup from each group using the MouseOx Pulse Oximeter and WinDaq Waveform Browser software (STARR Life Sciences Corp., Oakmont PA, USA) before and after IH exposure. This procedure was conducted to confirm that the animals were hypoxic during IH cycling.

### 2.4. Sample Collection & Processing

Twelve groups of rats were studied (*n* = 18 pups/group; 9 males and 9 females; 36 eyes/group), 6 groups in RA and 6 groups in IH. At euthanasia, eyes were enucleated and rinsed in ice-cold phosphate-buffered saline (PBS, pH 7.4) on ice. Enucleation was performed with the use of iris forceps and scissors for separation of the eyes from the surrounding connective tissue, nerves, and muscle. For assessment of ocular growth factors, 12 eyes were used (6 males and 6 females). Eyes were pooled for a total of 4 vitreous fluid, retinal, and choroidal samples (2 males and 2 females). The vitreous fluid was obtained by gently perforating the eyes and centrifugation at 5000 rpm at 4 °C for 15 min, the vitreous fluid was collected in a collection Eppendorf tube. Dissection and harvesting of the retinal and choroidal tissues were conducted as previously described [5,6,8,56,57,58]. Isolated retinas and choroids were placed in sterile Lysing Matrix D tubes (2.0 mL) containing 1.4 mm ceramic spheres (MP Biomedicals, Santa Ana, CA, USA) and 1.0 mL sterile normal PBS, snap-frozen in liquid nitrogen, and stored at −80 °C until analysis. For assessment of retinal vascular density, whole eyes (6 per group, 3 males, and 3 females) were placed in 4% paraformaldehyde (PFA) pH 7.4 for 90 min prior to flatmounting and adenosine diphosphatase (ADPase) staining of the superficial vasculature. Images were used for quantification of the retinal vessel diameter. For assessment of the retinal astrocyte and vascular integrity, whole eyes (6 per group, 3 males and 3 females) were flatmounted in ice-cold PBS pH 7.4 and stained for GFAP (astrocytes) and isolectin B4 (vasculature). For retinal histopathology and morphometry, whole eyes (6 per group, 3 males, and 3 females) were placed in 10% phosphate-buffered formalin and sent to the Pathology Department at SUNY Downstate Medical Center (Brooklyn, NY, USA) for processing, embedding, and H&E staining using standard laboratory techniques. Unstained sections were used for staining of HIF_1α_ and VEGF using immunohistochemistry. The remaining eyes were used for Western blot analyses of HIF_1α_ in the retina and choroid. Retinas and choroids were isolated and placed in sterile Lysing Matrix D tubes (2.0 mL) containing 1.4 mm ceramic spheres (MP Biomedicals, Santa Ana, CA, USA), snap-frozen in liquid nitrogen, and stored at −80 °C until analysis.

### 2.5. Assay of Growth Factors

All samples were analyzed on the same day. On the day of analyses, the tubes containing tissue in PBS were allowed to defrost on ice and were placed in a high-speed FastPrep-24 instrument (MP Biomedicals, Santa Ana, CA, USA), which utilizes a unique, optimized motion to efficiently homogenize biological samples within 40 seconds via multidirectional simultaneous beating of the Lysing Matrix ceramic beads on the tissue. This system prevents sample–sample contamination. The homogenates were then centrifuged at 4 °C at 10,000 rpm for 20 min. The supernatant was filtered, and the filtrate was used for the assays. A portion of the filtrate was used for total cellular protein levels. Vascular endothelial growth factor (VEGF), soluble VEGF receptor (sVEGFR)-1, sVEGFR-2, and insulin-like growth factor (IGF)-I levels were determined in the ocular samples (vitreous fluid, retina, and choroid) using commercially-available VEGF and IGF, VEGFR-1, and VEGFR-2 Quantikine enzyme-linked immunosorbent assay (ELISA) kits, respectively, purchased from R&D Systems (Minneapolis, MN, USA). All samples were processed and assayed according to the manufacturer’s protocol and the mouse VEGFR-1 ELISA kits detected rat VEGFR-1. Levels in the retinal and choroidal tissue homogenates were standardized using total cellular protein levels. 

### 2.6. Total Cellular Protein Levels

On the day of assays an aliquot (10 µL) of the retinal and choroid homogenates was utilized for total cellular protein levels using the Bradford method (Bio-Rad, Hercules, CA, USA) with bovine serum albumin as a standard.

### 2.7. Western Blots

All samples were analyzed on the same day. On the day of the assay, 400 µL ice-cold radioimmunoprecipitation assay (RIPA) lysis buffer was added to the tubes containing the retina and choroid tissue samples. The samples were homogenized in a high-speed FastPrep-24 instrument (MP Biomedicals, Santa Ana, CA, USA) as described above. After addition of 200 µL ice-cold RIPA lysis buffer to each tube, the samples were agitated for 2 hours at 4°C in an orbital shaker, then centrifuged at 12,000 rpm for 20 min at 4 °C. The protein content of the supernatant was determined using the BioRad protein assay and all samples were adjusted to 5 mg/mL protein. After addition of equal volumes of 2× Laemmli sample buffer containing 2-mercaptoethanol, the samples were boiled for 5 min at 95 °C, then loaded (15 µL) onto mini Protean TGX precast gels (BioRad, Hercules, CA, USA) for electrophoresis in 1× tris-glycine/SDS buffer, pH 8.3. The proteins were then transferred to Trans Blot transfer membranes using the Trans Blot Turbo machine (BioRad, Hercules, CA, USA), confirmed using Ponceau S solution. After blocking the membrane with 1% TBS with casein (BioRad, Hercules, CA, USA), HIF_1α_ primary antibody (1:10,000 dilution) purchased from Santa Cruz Biotechnology (Dallas, TX, USA) were added. The membranes were incubated with gentle agitation at 4 °C overnight and washed in tris-buffered saline with tween 20 (TBST) buffer prior to addition of horseradish peroxidase (HRP)-conjugated secondary antibodies (1:10,000 dilution) and incubation for 1 hour. The membranes were washed and the detection substrate consisting of luminol and peroxide (1:1 ratio) added. The membranes were imaged using the ChemiDoc Imaging system (BioRad, Hercules, CA, USA) and analyzed using the Image Lab software (ver. 4.1, BioRad, Hercules, CA, USA). 

### 2.8. Retinal Flatmounts

Eyes were enucleated and placed in 4% paraformaldehyde (PFA, pH 7.4) on ice. The eyes were placed in the refrigerator for 90 min, after which they were placed in ice-cold phosphate-buffered saline (PBS) on ice. Following removal of the cornea and lens, the retina was separated from the sclera, cut in 4 quadrants, and flattened. Retinas dedicated for ADPase staining were immersed in 4% PFA and stored overnight at 4°C. Retinas dedicated for glial fibrillary acidic protein (GFAP) and isolectin B4 double staining were immersed in PBS/Triton X-100 (TXPBS).

### 2.9. ADPase Staining of the Retinas

After 24 hours of incubation in 4% PFA, the retinas were washed in tris maleate buffer (pH 7.2) on ice prior to incubation in ADPase incubation medium containing 3.0 mM lead nitrate and 6.0 mM magnesium chloride (Sigma Chemical Co., St. Louis, MO) in tris maleate buffer (pH 7.2). After incubation, the retinas were washed with tris maleate buffer prior to addition of diluted ammonium sulfide (Fisher Scientific, Silver Spring, MD, USA) for 1 min. The retinas were washed again in tris maleate buffer and flat-mounted on a microscope slide with PBS and glycerin. All images were captured using an Olympus BX53 microscope, DP72 digital camera, and CellSens Dimension imaging software (version 2.1) from Olympus America, Inc. (Center Valley, PA, USA), attached to a HP Z44 computer. 

### 2.10. GFAP and Isolectin B4 staining

GFAP and isolectin B4 double staining of the retinal flatmounts were conducted as previously described [56,57,58]. Briefly, retinal flatmounts were washed in ice-cold PBS/Triton X-100 (TXPBS), fixed in methanol, followed by permeabilization and blocking in PermBlock (PBS + 0.3% Triton X-100 + 0.2% bovine serum albumin) in 5% goat serum for 1 hour. After washing in TXPBS, flatmounts were incubated with rabbit GFAP primary antibody (Cell Signaling Technologies, Danvers, MA, USA) overnight at 4°C. Following several washes with TXPBS, the flatmounts were incubated with Alexa Fluor 488 goat anti-rabbit fluorescent secondary antibodies, and Alexa Fluor 594 Isolectin B4 (ThermoFisher Sci/Life Technologies, Grand Island, NY, USA) overnight at 4 °C. The flatmounts were washed with TXPBS and mounted on slides with prolong anti-fade fluorescent mounting media and imaged at 20X magnification using the Olympus BX53 microscope, DP72 digital camera, and CellSens Dimension imaging software (version 2.1) from Olympus America, Inc. (Center Valley, PA, USA), attached to an HP Z44 computer.

### 2.11. Retinal Angiogenesis and Morphometric Analyses

Tortuosity index, vessel diameter, and number of endothelial cells present in the nerve fiber layer (NFL)/ganglion cell layer (GCL) were used to determine retinal angiogenesis, as previously described [56,57,58]. All measurements were conducted using the CellSens software (Olympus America, Inc. Center Valley, PA, USA). The diameter of the arteries and the veins was measured around the optic nerve from the optic disk to the first branch using the arbitrary line tool. Tortuosity of the vessels was quantified by tracing a line along the tortuous vessel using the polyline tool and comparing the length of the tortuous vessel to the length of the straight vessel from the optic disk to the first branch using the arbitrary line tool. 

Vascular density reflective of vasoproliferation was determined using the polygon tool and count and measure on region of interest, and is presented as a percent of the total area of interest. The number of endothelial cells present in a defined area of 1000 µm was determined using the count and measure tool. Measurements for central corneal thickness, total retinal thickness, and thickness of each retinal layer corresponding to the nerve fiber layer/ganglion cell layer (NFB/GCL), inner plexiform layer (IPL), inner nuclear layer (INL), and outer nuclear layer (ONL) were used for morphometric analyses using the arbitrary line tool. 

### 2.12. Histopathology

Whole eyes were fixed in-situ in 10% neutral buffered formalin (NBF), then enucleated, marked for identification, placed in cassettes, and immersed in 10% NBF. The samples were taken to the State University of New York (SUNY) Downstate Medical Center Pathology Department for processing, embedding, sectioning, staining, and mounting using standard laboratory techniques. Unstained sections were used for immunohistochemistry.

### 2.13. Immunohistochemistry

Formalin-fixed tissue sections (5 microns) were deparaffinized in a series of xylenes and graded alcohols, followed by washing in de-ionized water. After unmasking the antigens using sodium citrate buffer, pH 6.0, sections were washed in PBS and incubated in a humidified chamber for 1 hour at room temperature in 5% normal blocking serum in PBS. After removal of the blocking serum, HRP-conjugated primary antibodies (Santa Cruz Biotechnology, Inc. Dallas, TX, USA) diluted in PBS with 5% normal blocking serum (2 µg/mL) was added to the sections prior to incubation overnight at 4°C. After washing in PBS, a mixture of hydrogen peroxide (H_2_O_2_) and 3,3′-diaminobenzidine (DAB) diluted in 0.1 M Tris-HCl pH 7.6 was added to the section until development of color. The sections were washed in deionized H_2_O and counterstained with hematoxylin followed by dehydration in ethanol and xylene, then mounted with SignalStain mounting medium (Cell Signaling, Danvers, MA, USA). Images were captured at 40X magnification using the Olympus BX53 microscope, DP72 digital camera, and CellSens Dimension imaging software (version 2.1) from Olympus America, Inc. (Center Valley, PA, USA), attached to a HP Z44 computer. 

### 2.14. Statistical Analysis

Differences among the RA and IH groups, and differences among the supplemental groups within each oxygen environment were analyzed using two-way analysis of variance (ANOVA) with Dunnett’s post-hoc tests, following the Bartlett’s test for normality. Kruskal-Wallis nonparametric test with Dunn’s multiple comparison was used for non-normally distributed data. The percentage of retinas that scored >18 in each group was calculated and analyzed using the Fisher’s exact test. Data were analyzed using the IBM SPSS Statistics software, version 26.0 (SPSS Inc., Chicago, IL, USA) and are reported as mean ± standard deviation (SD). A *p*-value of <0.05 was considered as statistically significant.

## 3. Results

### 3.1. Growth

Low birth weight and poor postnatal growth are important predictors of ROP. Percentage change in body weight and body length from birth were determined at P7, P14, and P21 in order to establish the effects of oral supplementation and/or topical ocular ACV on anthropometric growth. Data are presented in Table 1. In the RA groups, oral nGSH and topical ocular PBS consistently resulted in the highest weight accretion at all time intervals. This effect was abolished when nGSH was combined with topical ocular ACV, resulting in the lowest weight gain at P14 and P21. Although *n*-3 PUFAs + ACV resulted in the lowest weight gain at P7, the animals exhibited modest catchup growth at P14 and P21 and surpassed that of the nGSH + ACV group, but was lower than the other groups. This same group also exhibited the least linear growth in RA, suggesting increased body fat and/or edema. In IH, a quite different pattern emerged. Highest weight gain was achieved with CoQ10 + ACV at P7, however by P14 and P21, highest weight gain was achieved with CoQ10 + *n*-3 PUFAs + ACV. In contrast, *n*-3 PUFAs + ACV resulted in consistently low weight gain throughout the entire experimental time. Interestingly, nGSH + PBS resulted in the greatest linear growth over the 3-week period, while *n*-3 PUFAs + ACV resulted in the lowest.

### 3.2. Organ/Body Weight Ratios

Organ weights are important endpoints in toxicology studies. Due to differences in body weight, analysis of organ-to-body weight ratios normalizes weight differences and represents important indices of disproportional vital organ weight gain or loss in response to treatment. Data showed that in RA, the highest brain/body weight ratios were achieved with nGSH + ACV (0.037 ± 0.001) and CoQ10 + *n*-3 PUFAs + PBS (0.037 ± 0.001) compared to all other groups (0.03 ± 0.001). In IH, the highest brain/body ratios were achieved with nGSH + PBS (0.04 ± 0.002), nGSH + ACV (0.04 ± 0.001), and *n*-3 PUFAs + ACV (0.04 ± 0.001) compared to the other groups, with the lowest being CoQ10 + *n*-3 PUFAs + ACV (0.035 ± 0.001). Of the RA groups, lung/body weight ratios were lowest in the nGSH + PBS (0.009 ± 0.0005) compared to the other groups (0.01 ± 0.0005). In IH, lung/body weight ratios were highest in the nGSH + ACV (0.014 ± 0.0007) and *n*-3 PUFAs + ACV (0.014 ± 0.0006) groups compared to the other groups (0.01 ± 0.0060). Liver/body weight ratios were lowest in the nGSH + PBS group (0.037 ± 0.002) compared to the other groups (0.04 ± 0.002) in RA, while in IH, nGSH + ACV (0.037 ± 0.002) and CoQ10 + *n*-3 PUFAs + PBS (0.037 ± 0.001) resulted in the lowest liver/body weight ratios compared to the other groups (0.04 ± 0.002). No significant differences were noted among the groups for heart/body weight ratios or kidney/body weight ratios in RA or IH.

### 3.3. Eye Opening

The retina of rats develops and matures postnatally. At birth, the rat retina is largely undifferentiated and by P14 (the time of eye opening) many retinal cells are mature and the layers are differentiated. The period from conception to eye opening is called the “caecal period” during which maturation of the retinal neural circuitry, corneal development, and overall maturation of the visual cortex occurs. We determined whether oral supplementation and topical NSAIDs have synergistic benefits for precocious retinal neural circuitry development. Eyes were assessed daily for signs of opening and the results are presented in Table 2. We observed that 16 of the 18 rats (89%) in the nGSH + ACV group exposed to RA had both eyes open on P13 and by P14, 100% were open. This effect was drastically disrupted in IH, resulting in only 61% open at P14 (*p* < 0.01 vs. RA). Of the RA groups, *n*-3 PUFAs + ACV resulted in the least number of opened eyes at P14 (14%) compared to nGSH + ACV (100%). Of the IH groups, all groups had reduced eye opening compared to their RA counterparts, but of all groups, CoQ10 + *n*-3 PUFAs + ACV resulted in the greatest number of opened eyes (89%) whereas the nGSH + PBS resulted in the least (33%). 

### 3.4. Ocular HIF_1α_ Levels

Ischemic conditions strongly induce the hypoxia-inducible transcription factor (HIF) complex, which binds to a hypoxia response element within the VEGF promoter, mediating the angiogenic response that follows ischemic tissue injury. HIF_1α_ levels in the retina and choroid were determined using Western blot analyses. Figure 1 shows that treatment with combined *n*-3 PUFAs + ACV, as well as CoQ10 + *n*-3 PUFAs with or without ACV, eliminated the expression of HIF_1α_ in the retina and choroid when treatment was administered in RA and IH. 

### 3.5. Systemic and Ocular VEGF

Because pups received oral supplementation and topical ocular treatments, it was important to determine the systemic and ocular effects on the angiogenesis factors. Systemic and ocular VEGF levels are presented in Figure 2. The first observation was that systemic VEGF levels were 2-fold lower than ocular levels and of the ocular levels, retinal VEGF was lowest. Of the RA groups, serum levels (panel A) were lowest with nGSH + ACV and highest with CoQ10 + *n*-3 PUFAs with or without ACV. Of the IH groups, serum levels were lowest with *n*-3 PUFAs + ACV and CoQ10 + *n*-3 PUFAs + ACV and highest in the nGSH + PBS group compared to RA and all other IH groups. In the vitreous fluid (panel B), *n*-3 PUFAs + Acuvail reduced VEGF levels in RA and IH compared to all other groups. In RA, nGSH + PBS produced the highest VEGF levels, whereas in IH, CoQ10 + ACV produced the highest. Retinal VEGF levels (panel C) were also lower with *n*-3 PUFAs + Acuvail in IH, but highest levels were measured in the nGSH + ACV in IH. In the choroid (panel D), VEGF levels increased substantially in the nGSH + PBS and *n*-3 PUFAs + ACV groups exposed to IH. Treatment with nGSH + ACV, and CoQ10+ *n*-3 PUFAs with and without ACV suppressed choroidal VEGF levels in RA and IH.

### 3.6. Systemic and Ocular sVEGFR-1

Soluble VEGFR-1 is a splice variant of the membrane VEGFR-1 receptor which acts as a VEGF “trap”, making VEGF less available to its membrane receptors and thus reducing VEGF action. Levels are presented in Figure 3. Similar to VEGF, serum levels were approximately 2-fold lower compared to the ocular compartment. In the serum (panel A), combined CoQ10 + *n*-3 PUFAs + ACV substantially induced sVEGFR-1 in IH, while *n*-3 PUFAs + ACV resulted in the lowest levels. In the vitreous fluid (panel B), *n*-3 PUFAs + ACV treatment also resulted in the lowest levels in RA and IH. In RA, CoQ10 + *n*-3 PUFAs + PBS produced the highest levels, while in IH, CoQ10 + ACV were highest. In the retina (panel C), the highest levels were noted with *n*-3 PUFAs in RA and lowest levels were noted with the same treatment in IH. In the choroid, the highest RA levels were noted with CoQ10 + ACV and the lowest with *n*-3 PUFAs + ACV and CoQ10 + *n*-3 PUFAs + ACV, while in IH, choroidal sVEGFR-1 levels were the highest with nGSH + PBS and the lowest with CoQ10+ *n*-3 PUFAs + ACV.

### 3.7. Systemic and Ocular sVEGFR-2

Soluble VEGFR-2 is a splice variant of its membrane type receptor and like its sVEGFR-1 counterpart, acts as a negative regulator of VEGF by trapping VEGF. Levels are presented in Figure 4. It was interesting to note that sVEGFR-2 was not detected in the vitreous fluid. In contrast to VEGF and sVEGFR-1, sVEGFR-2 levels were 10-fold higher in the serum compared to the retina and choroid. Of further interest was the differential suppressive effect of *n*-3 PUFAs + ACV and CoQ10 + ACV on serum sVEGFR-2 (panel A) in IH conditions compared to RA. This effect persisted in the retina (panel B) with *n*-3 PUFAs + ACV only. Of note, the group treated with combined CoQ10 + ACV and CoQ10 + *n*-3 PUFAs with and without ACV had higher sVEGFR-2 levels in IH compared to RA. In the choroid (panel C), the highest sVEGFR-2 levels were attained with CoQ10 + ACV treatment in RA compared to all other RA groups, while among the IH groups, nGSH + PBS and *n*-3 PUFAs + ACV were highest.

### 3.8. Systemic and Ocular IGF-I

IGF-I is a permissive factor for VEGF and low serum levels have been shown to be predictive for severe ROP. Levels are presented in Figure 5. IGF-I levels in the serum were significantly higher compared to the ocular compartment. In the ocular compartment, vitreous fluid levels were highest, while retinal levels were lowest. In the serum (panel A), nGSH + PBS treatment in RA resulted in the highest IGF-I levels compared to all other RA groups. In IH, only *n*-3 PUFAs + ACV H caused a significant decline in IGF-I. In the vitreous fluid, no major changes were noted among the RA groups, but in IH, IGF-I levels declined with *n*-3 PUFAs + ACV and CoQ10+ *n*-3 PUFAs + ACV. In the retina, *n*-3 PUFAs + ACV resulted in the highest levels in RA, but the lowest levels in IH. Only nGSH + ACV caused higher IGF-I levels in IH. Similarly, in the choroid (panel D), nGSH + PBS resulted in elevated IGF-1 levels in IH compared to all other IH groups, which were lower or unchanged from their RA counterparts. Of the IH groups, CoQ10+ *n*-3 PUFAs + ACV resulted in the lowest choroidal IGF-I levels. 

### 3.9. Corneal Thickness and Histopathology

Central corneal thickness measurements are associated with intraocular pressure. Since topical ocular ACV administration started at P5 when the eyelids were fused, we assessed the corneas for adverse effects such as abrasions and thickness. Corneal histopathology is presented in Figure 6 (images are 40X magnification, scale bar is 20 µm). The RA groups are panels A–F (upper) and the IH groups are panels G–L (lower). In RA, treatment with nGSH + PBS (panel A) nGSH + ACV (panel B) and *n*-3 PUFAs + ACV (panel D) caused moderate corneal thickening (142.6 ± 5.9, 132.2 ± 2.1 and 143.6 ± 2.9, respectively), while corneal thinning was observed with CoQ10 + ACV (panel C, 112.2 ± 1.6) and CoQ10 + *n*-3 PUFAs + ACV (panel F, 126.3 ± 1.9). In IH, nGSH + ACV (panel H) resulted in thick disrupted widely separated collagen fibers of the stroma (158.4 ± 0.99). Thickness of the corneas from the CoQ10 + ACV (panel I), *n*-3 PUFAs + ACV (panel J), and CoQ10 + *n*-3 PUFAs + ACV groups did not appreciably differ from RA. However, treatment with CoQ10 + *n*-3 PUFAs + PBS (panel K) resulted in corneal thinning (111.1 ± 1.09).

### 3.10. Retinal Vasculature

Representative ADPase stained retinas from the RA and IH groups are presented in Figure 7 and Figure 8, respectively. Images are 10× magnification (scale bar is 100 µm). Composite images for each group include retinal vasculature at the periphery (upper, numbered 1) and retinal vasculature at the optic disk (lower, numbered 2). Figure 7 shows no significant abnormalities in the retinal vasculature at the periphery or optic disk with each treatment in RA. In contrast, exposure to IH (Figure 8) showed the presence of vascular tufts, tortuous vessels, hemorrhage, and abundant disorganized vasculature. Treatment with nGSH + PBS (panels A1 and A2) show tortuous, disorganized vessels and vascular tufts at the periphery (arrows), but no adverse outcomes at the optic disk. In contrast, treatment with nGSH + ACV (panels B1-B4) shows vascular tortuosity, hemorrhage, and enlarged vessels (arrows) at the periphery and optic disk. Treatment with CoQ10 + ACV (panels C1 and C2) show tortuous enlarged vessels at the periphery (arrow), but no adverse outcomes at the optic disk. Treatment with *n*-3 PUFAs + ACV (panels D1-D4) shows tortuous, enlarged vessels with punctate hemorrhages (arrows) at the periphery, and enlarged tortuous vessels with hemorrhage around the optic disk (arrows). Treatment with CoQ10 + *n*-3 PUFAs + PBS (panels E1-E4) show similar enlarged tortuous vessels with hemorrhage at the periphery and optic disk (arrows), but to a lesser extent. Treatment with CoQ10 + *n*-3 PUFAs + ACV (panels F1 and F2) provided the most benefit with only mild hemorrhage at the periphery (arrow). 

Analysis of the vascular density of the periphery vessels showed significant elevations in the nGSH + ACV (11.4 ± 2.9 vs. 9.2 ± 2.9, *p* < 0.01) and CoQ10 + ACV (10.6 ± 3.4 vs. 7.5 ± 2.1, *p* < 0.01) RA vs. IH groups. Within each oxygen environment, *n*-3 PUFAs + ACV exhibited the highest vascular density compared to the other groups (RA: 12.2 ± 2.4 and IH: 12.6 ± 3.5, *p* < 0.01), followed by nGSH + ACV in IH (11.4 ± 2.9, *p* < 0.01). Morphometric analyses (Table 3) show that treatment with *n*-3 PUFAs + ACV and CoQ10 + *n*-3 PUFAs + PBS in IH exhibited the highest vessel tortuosity index and arterial diameter compared to the other groups. 

### 3.11. Retinal Astrocytes

Representative GFAP (green)/isolectin B4 (red) stained merged images of retinas from the RA and IH groups are presented in Figure 9 and Figure 10, respectively. Images are 10× magnification (scale bar is 100 µm). The upper panels (numbered 1) are images from the periphery and the lower panels (numbered 2) are images from the optic disk. In the RA groups (Figure 9), there were no significant adverse effects on the retinal astrocytic template, except for mild disturbances at the periphery (arrows) in the *n*-3 PUFA + ACV (panels D1 and D2) and CoQ10 + *n*-3 PUFAs + ACV (panels F1 and F2). In contrast, retinas exposed to IH (Figure 10) showed major disturbances in the astrocytic template. Groups treated with nGSH/PBS (panels A1 and A2) and nGSH/ACV (panels B1 and B2) showed mild disturbances at the periphery (arrows). In contrast, groups treated with CoQ10 + ACV (panels C1 and C2), *n*-3 PUFAs + ACV (panels D1 and D2), and CoQ10 + *n*-3 PUFAs + PBS (panels E1 and E2), showed significant astrocyte disruptions at the periphery and optic disk (arrows), and possible Müller end foot reactivity. In contrast, combined CoQ10 + *n*-3 PUFAs + ACV (panels F1 and F2), resulted in mild disturbances at the periphery and decreased astrocyte reactivity for GFAP, with no adverse outcomes at the optic disk. 

### 3.12. Retinal Layers

Representative H&E stained retinal layers for groups in RA and IH are presented in Figure 11. Images are 40× magnification (scale bar is 20 µm). The RA groups are panels A–F and the IH groups are panels G–L. No major abnormalities were noted with the treatments in RA (panels A–F). However, significant choroidal hemorrhage and major abnormalities such as rosettes were seen in the INL and ONL layers (arrows) in the IH groups treated with *n*-3 PUFAs + ACV (panels J1 and J2) and CoQ10 + *n*-3 PUFAs + PBS (panels K1 and K2). Morphometric analyses (Table 3) shows significantly less endothelial cells in the NFL/GCL layer of the groups treated with CoQ10 + *n*-3 PUFAs + ACV consistent with Figure 11 (panels F and L). Overall retinal thickness and NFL/GCL layers were also reduced compared to the other IH groups.

## 4. Discussion

The present study is a follow-up to our previous reports which showed that topical ocular ACV [7], and oral CoQ10 or *n*-3 PUFAs given individually [8], attenuated the severity of IH-induced OIR, but did not entirely prevent or eliminate the unwanted vascular pathologies consistent with severe ROP. Each treatment demonstrated mechanistically distinct beneficial attributes, targeting inflammation, oxidants, or promoting growth, leading us to hypothesize that combining the treatments would lead to synergistic enhancement and optimization of their particular attributes. Since the etiology of OIR/ROP is complex and involves numerous mechanisms associated with oxidative stress, poor growth, and inflammation, we utilized a blend of oral antioxidant/anti-inflammatory/growth-promoting supplements to a non-invasive, topical ocular NSAID previously shown to be safe and effective in preterm infants for ROP [55], and in our OIR model [7]. While the mechanism of NSAID inhibition of angiogenesis is complex, studies show that NSAIDs have inhibitory effects on HIF_1α_ and von-Hippel-Lindau (VHL) [59], and exogenous prostaglandins partly reverses NSAID inhibition of angiogenesis. These inhibitory effects may be due, at least in part, to the strong upregulation of COX-2 and PGE_2_ and subsequently, VEGF, by hypoxia [51,52,60]. Our data showed that the most efficacious pharmacologic blend for simultaneous reduction of biomarkers of oxidative stress, angiogenesis, and inflammation, and preservation of retinal architecture, was oral CoQ10 + *n*-3PUFAs with topical ocular ACV. nGSH + PBS also showed significant benefits for reducing OIR, but adverse effects on postnatal growth persisted. The positive effect of singular nGSH may indicate that lipid peroxidation (via accumulation of H_2_O_2_) is highly involved in OIR and supports our previous work [56]. This approach of combining mechanistically distinct therapies with delivery methods to synergize and optimize therapeutic benefits may have promising clinical applications. 

The effects of any new treatment on organ weight is important in toxicology experiments. Organ weight is considered one of the most sensitive indicators of an effect of an experimental compound [61]. Due to differences in weight at birth and the presence of runts in some litters, we used the ratio of organ weight to body weight in order to eliminate those differences. In addition, monitoring postnatal weight gain is a key non-invasive tool for identifying ROP risk [62]. Singular nGSH supplementation in RA outperformed the other supplements with respect to growth accretion (body weight and length) at all time points, and resulted in the highest serum IGF-I levels. The effect of nGSH on weight has been previously shown in pediatric patients [63]. However, one of the most disturbing findings was the deleterious effect of IH on nGSH growth promotion. While this phenomenon has not been appreciably investigated, particularly in ELGANs who experience frequent IH and apneic events during the first few weeks of life, there are a few studies that have shown that IH impairs drug metabolism [64,65,66], suggesting reduced efficacy. This may have a significant impact on pharmacokinetics in ELGANs who are exposed to numerous drugs due to multiple morbidities. Interestingly, all IH groups exhibited lower IGF-I levels compared to their RA littermates. Nevertheless, in the case of nGSH, it was reassuring that linear growth was not affected, and suggests either a reduction in body fat and/or excess water. This same group demonstrated good retinal outcomes compared to some of the other combination treatments, despite IGF-I levels. In addition, regardless of higher IGF-I levels in the vitreous fluid, retinal outcomes were not improved. Of the other groups, *n*-3 PUFAs + ACV resulted in the lowest weight gain and linear growth. This was reflected in the serum IGF-I levels and may suggest that combining *n*-3 PUFAs with NSAIDs is detrimental to growth. Of all organs, the brain, liver, and lungs were most affected by the supplements. Consistent with our previous findings [8], CoQ10 and *n*-3 PUFAs, as well as nGSH, may spare those organ weights when exposed to chronic neonatal IH. None of the treatment combinations in RA or IH significantly reduced organ/body weight ratios below saline and/or RA control levels.

Another interesting finding was the effect of nGSH with and without ACV on precocious eye opening when administered in RA, an effect that was disrupted with IH. As stated above, we observed that 72% of pups in the nGSH RA group opened both eyes on P13 with 100% open on P14. At birth (P0), the rat retina is largely undifferentiated and by P14 (eye opening), many retinal cells are mature. The period from conception to eye opening in rodents represents a time of maturation of the retinal neural circuitry, corneal development, and overall visual cortex development [67,68,69]. This precocious eye opening induced by nGSH suggests early maturation of the visual cortex, and was linked to improved somatic growth. Two important synaptic interactions occur at the time of eye opening, the splitting of the visual signal into two channels for light and dark objects and the instillation of pathways to create simultaneous contrast of visual objects [70]. Our findings support previous reports and provide further evidence for a role of GSH in retinal development [71]. Exposure to IH substantially delayed eye opening in a large percentage of rats in all groups (28–67%), and, in the case of nGSH, only 33% opened their eyes on P14, suggesting delayed retinal neural circuitry and visual cortex maturation. Clearly, this was an effect of IH. 

Ocular drug delivery is challenging, particularly when the target is posterior as in the case of the retina. Administration of drugs via systemic or topical ocular routes encounters many barriers. Systemic drugs are restricted by the blood-aqueous and blood-retinal barriers. However, drugs can enter into the choroid, but the retinal pigmented epithelial (RPE) restricts entry from the choroid to the retina [72]. Topical ocular drugs are restricted by the corneal and conjunctival layers. For these reasons, we assessed two delivery methods: 1) GSH nanoparticles with the assumption that nanoparticles would easily pass through the blood-retinal barrier due to instant absorption; and 2) oral combined with topical ocular delivery in an effort to optimize efficiency and reduce barrier limitations. To assess efficacy, we employed several techniques including angiogenesis biomarkers, retinal vascular and astrocyte integrity, retinal histopathology, morphometric analyses, and immunoreactivity and localization of angiogenesis biomarkers. Our approach of combination treatments and combination delivery methods resulted in unequivocal beneficial outcomes with the use of the oral CoQ10 + *n*-3 PUFAs blend co-administered with topical ocular NSAIDs, evidenced within the boundaries of the techniques employed herein. Retinal vascular and astrocytic integrity was preserved with no histopathological outcomes, to a much greater extent than the other treatment combinations. 

Identification of the ocular angiogenesis biomarkers associated with the most beneficial outcomes revealed reductions in retinal and choroidal HIF_1α_, VEGF, and sVEGFR-1, reductions in choroidal IGF-I, and elevations in retinal VEGFR-2. It was interesting to note the differences in responses between the retina and choroid. These differences are likely due to the structure and function of the two vascular systems. Unlike the retina, the choroid is fenestrated (leaky) and it is highly vascularized with a relatively high blood flow. A high VEGF gradient in the choriocapillaries is important for nurturing cone function [73]. Furthermore, since the cornea is continuous with the sclera, topical drug delivery may result in drug absorption through the conjunctiva, sclera, and choroid [74]. The VEGFRs, VEGFR-1, and VEGFR-2, are critical for VEGF signaling, and are the major mediators of proliferation, migration, and differentiation [75,76]. The affinity of VEGF for VEGFR-2 is at least 10-fold lower than its affinity for VEGFR-1 [77]. Soluble forms of VEGFRs trap VEGF, making it less available to signal to its membrane receptors. The response of sVEGFR-2 in the CoQ10 + *n*-3 PUFAs + ACV group, which produced the best outcomes, is interesting, increasing in the retina. This may suggest that sVEGFR-2 may be more important than sVEGFR-1 as an anti-angiogenic factor in this setting. Of all VEGFRs, sVEGFR-2 binds all forms of VEGF, except VEGF B. It must be mentioned that nGSH + PBS also showed similar beneficial efficacy, but its deleterious effects on postnatal growth requires further assessment. Whether combining nGSH with CoQ10 or *n*-3 PUFAs would impact on postnatal growth trajectories remains unknown. Nevertheless, the therapeutic benefits achieved within the context of the retina suggest that nGSH may have overcome some of the critical barriers associated with oral drug administration. The findings also suggest that oxidative stress and oxidants, particularly peroxides, which are detoxified by GSH, may play a key role in the development of severe OIR. An unexpected finding was the lack of efficacy of the other combination, as well as the reduced efficacy of nGSH + ACV. In particular, the combination of *n*-3 PUFAs + ACV, which resulted in many punctate hemorrhages (Figure 8), disrupted astrocyte template (Figure 10), choroidal hemorrhage, and ONL damage (Figure 11). Similar to adverse effects on growth, these outcomes strongly imply that combining *n*-3 PUFAs with NSAIDs is detrimental. The higher risk of hemorrhage with this combination may be due to enhanced antiplatelet effects. ACV was administered during a time when the eyelids were fused. Therefore, the risk of drug loss was diminished. Since newborn corneal membranes are thin compared to adults, drug absorption and corneal absorption is increased. 

## 5. Conclusions

Given the complex etiology of ROP, there is a critical need for identification and evaluation of more effective therapy and delivery approaches. No single treatment has proven successful without adverse short- or long-term adverse outcomes. Combination therapies and combination delivery methods that mechanistically synergize to optimize their distinct and individual efficacy attributes, target multiple underlying causes, and reduce barrier limitations, may prove to be a more rational approach. Here we show, for the first time, that combining oral CoQ10 + *n*-3 PUFAs and topical ocular NSAIDs simultaneously targeting oxidative stress, inflammation, and poor growth, mitigates the damaging effects of neonatal IH on the developing retina. Combining these therapies and delivery modalities enhance their individual benefits and simultaneously target multiple underlying pathways. This approach of not only combining treatments, but also of combining delivery methods, has broader clinical implications. 

## Figures and Tables

**Figure 1 nutrients-12-01980-f001:**
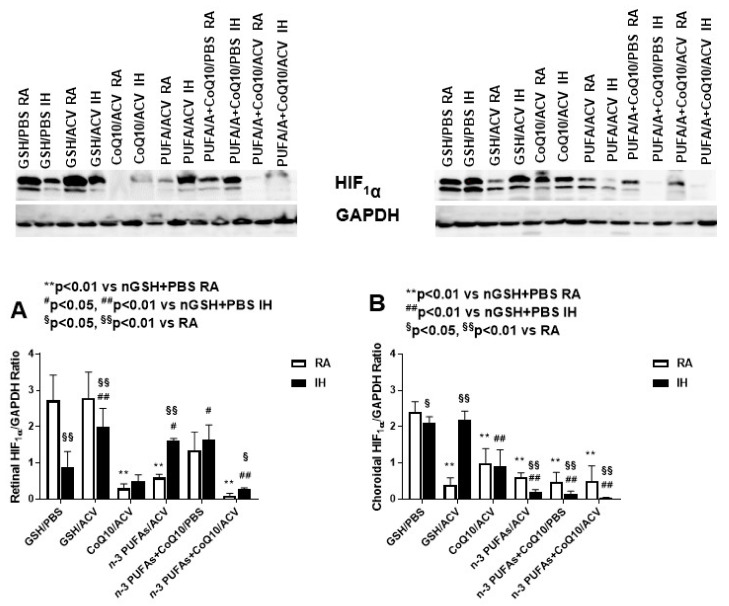
Expression of HIF_1α_ in the retina and choroid in response to combination treatments in RA and IH by Western blot analysis. Data are presented as mean ± SEM ratio of HIF_1α_/GAPDH (*n* = 4 replicates). Comparison among the RA or IH groups was conducted using one-way ANOVA with Bonferroni post hoc multiple comparisons test. Comparison between RA and IH for each group was conducted using unpaired *t*-test. Retinal HIF_1α_/GAPDH ratios (**A**), Choroidal HIF_1α_/GAPDH ratios (**B**).

**Figure 2 nutrients-12-01980-f002:**
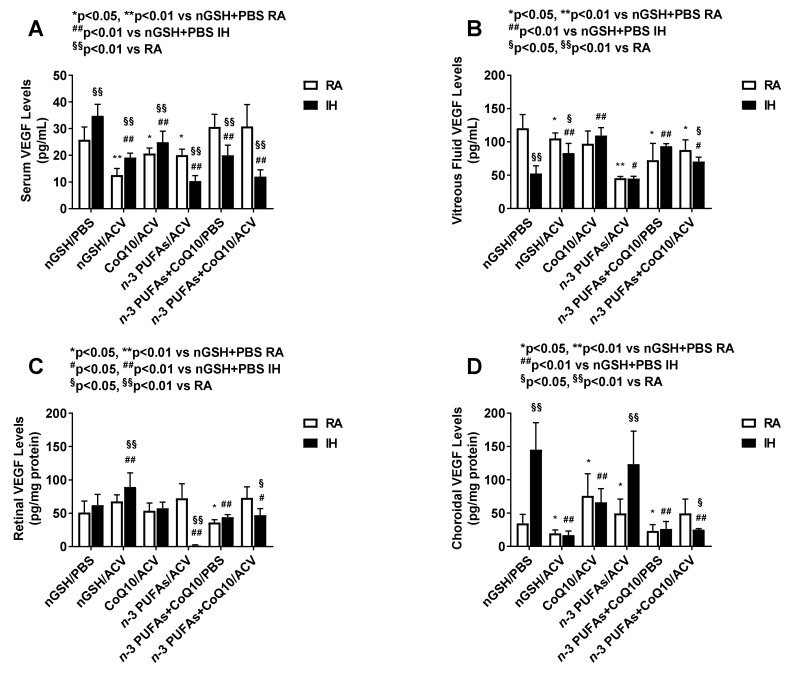
Effects of combination treatments on vascular endothelial growth factor (VEGF) levels in the serum (**A**), vitreous fluid (**B**), retina (**C**), and choroid (**D**). Levels in the retinal and choroidal homogenates were standardized using total cellular protein levels. The open bar represents the room air (RA) groups and the solid bar represents the IH groups. Data are expressed as mean ±SD (*n* = 6 samples/group for serum; and *n* = 4 samples/group for vitreous fluid, retina and choroid).

**Figure 3 nutrients-12-01980-f003:**
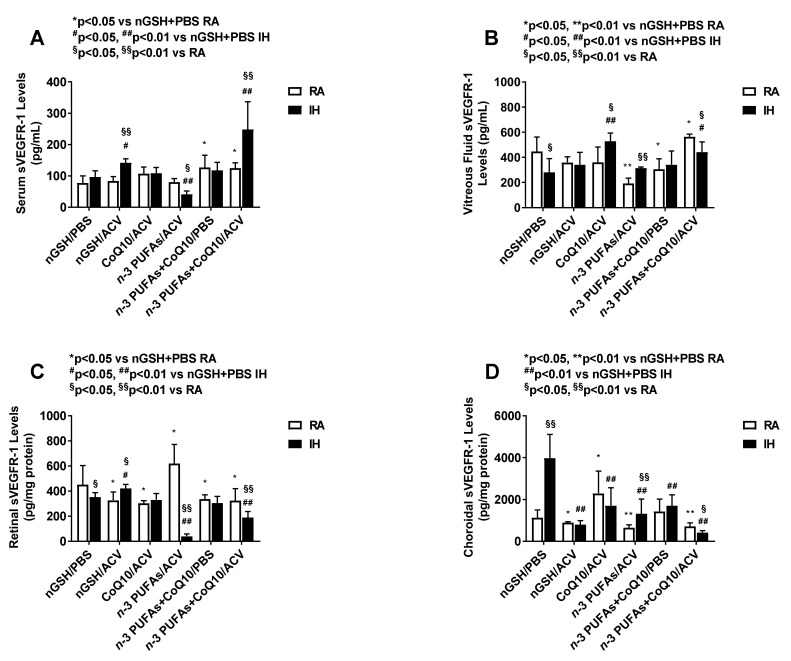
Effects of combination treatments on soluble vascular endothelial growth factor receptor (sVEGFR)-1 levels in the serum (**A**), vitreous fluid (**B**), retina (**C**), and choroid (**D**). Levels in the retinal and choroidal homogenates were standardized using total cellular protein levels. The open bar represents the room air (RA) groups and the solid bar represents the IH groups. Data are expressed as mean ±SD (*n* = 6 samples/group for serum; and *n* = 4 samples/group for vitreous fluid, retina and choroid).

**Figure 4 nutrients-12-01980-f004:**
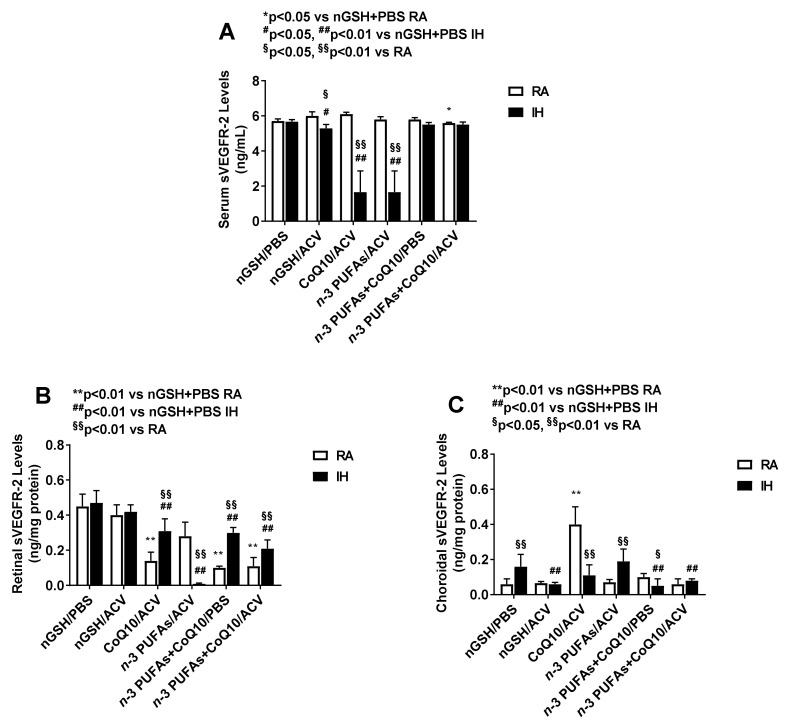
Effects of combination treatments on soluble vascular endothelial growth factor receptor (sVEGFR)-2 levels in the serum (**A**), retina (**B**), and choroid (**C**). sVEGFR-2 was not detected in the vitreous fluid. Levels in the retinal and choroidal homogenates were standardized using total cellular protein levels. The open bar represents the room air (RA) groups and the solid bar represents the IH groups. Data are expressed as mean ±SD (*n* = 6 samples/group for serum; and *n* = 4 samples/group for vitreous fluid, retina and choroid).

**Figure 5 nutrients-12-01980-f005:**
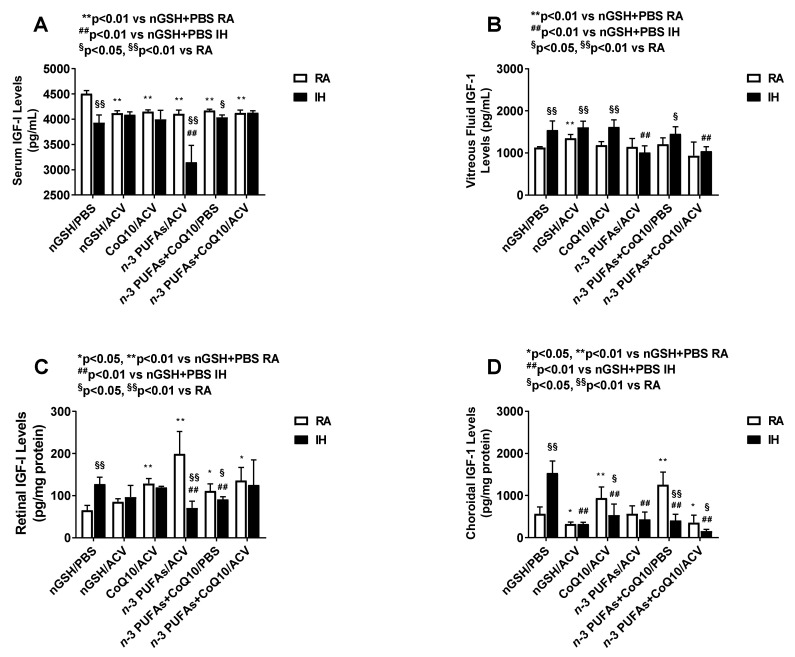
Effects of combination treatments on insulin-like growth factor (IGF)-I levels in the serum (**A**), vitreous fluid (**B**), retina (**C**), and choroid (**D**). Levels in the retinal and choroidal homogenates were standardized using total cellular protein levels. The open bar represents the room air (RA) groups and the solid bar represents the IH groups. Data are expressed as mean ±SD (*n* = 6 samples/group for serum; and *n* = 4 samples/group for vitreous fluid, retina, and choroid).

**Figure 6 nutrients-12-01980-f006:**
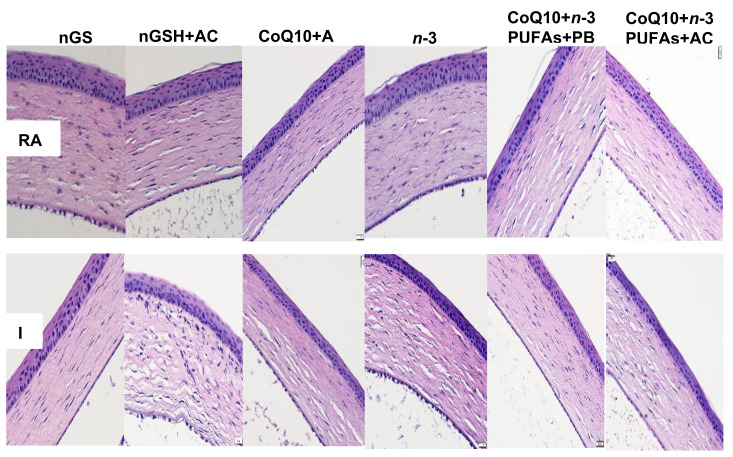
Representative H&E stained corneas from neonatal rats on postnatal day 21 (P21). Room air (RA) groups are represented in the top panels and intermittent hypoxia (IH) groups are represented in the bottom panels. Images are 40× magnification, scale bar is 20 µm.

**Figure 7 nutrients-12-01980-f007:**
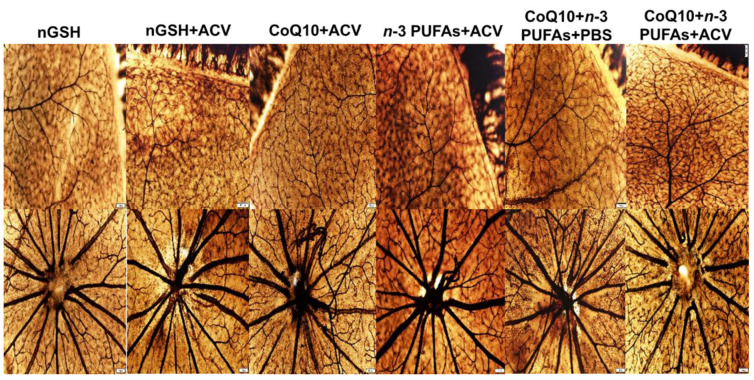
Representative ADPase-stained retinas from room air (RA) raised neonatal rats at postnatal day 21 (P21). The upper panels represent the periphery and the lower panels represent the optic disk. Images are 10× magnification, scale bar is 100 µm.

**Figure 8 nutrients-12-01980-f008:**
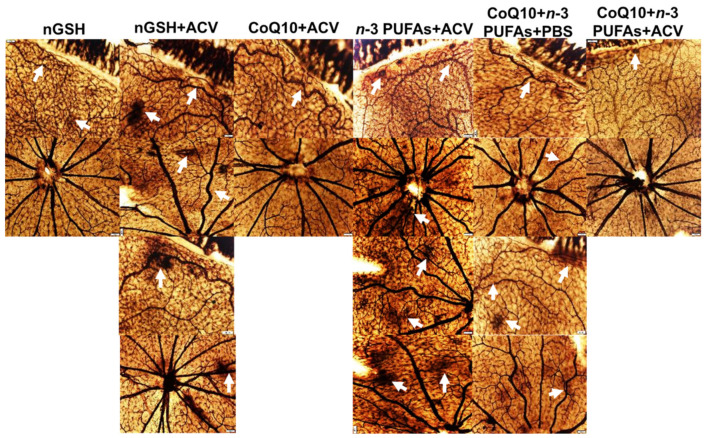
Representative ADPase-stained retinas from rats exposed to neonatal intermittent hypoxia (IH) at postnatal day 21 (P21). The upper panels represent the periphery and the lower panels represent the optic disk. Images are 10× magnification, scale bar is 100 µm. Arrow heads show retinal damage.

**Figure 9 nutrients-12-01980-f009:**
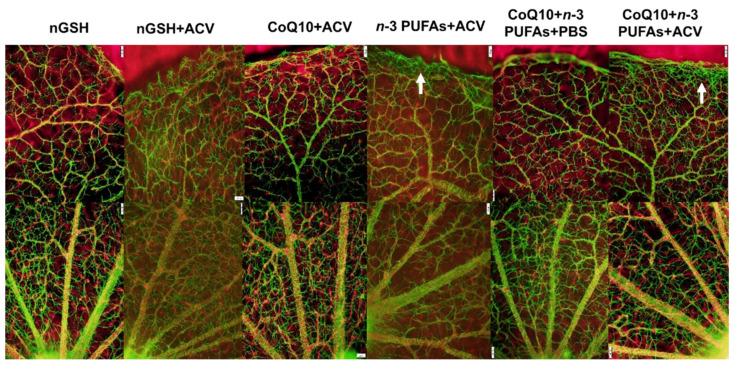
Representative glial fibrillary acidic protein (GFAP) (green)/isolectin B (red) stained merged images of retinas from room air (RA) raised neonatal rats at postnatal day 21 (P21). The upper panels represent the periphery and the lower panels represent the optic disk. Images are 10× magnification, scale bar is 100 µm.

**Figure 10 nutrients-12-01980-f010:**
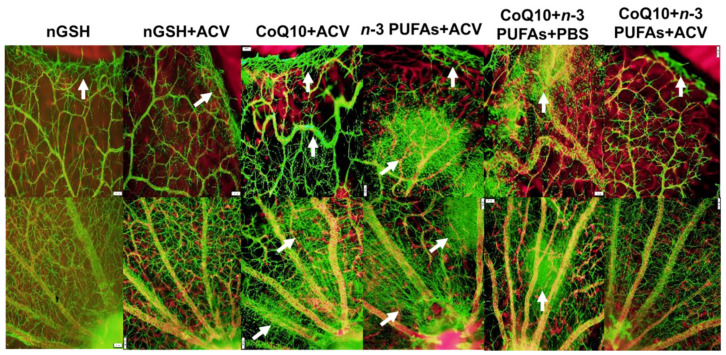
Representative GFAP (green)/isolectin B (red) stained merged images of retinas from rats exposed to neonatal intermittent hypoxia (IH) at postnatal day 21 (P21). The upper panels represent the periphery and the lower panels represent the optic disk. Images are 10× magnification, scale bar is 100 µm.

**Figure 11 nutrients-12-01980-f011:**
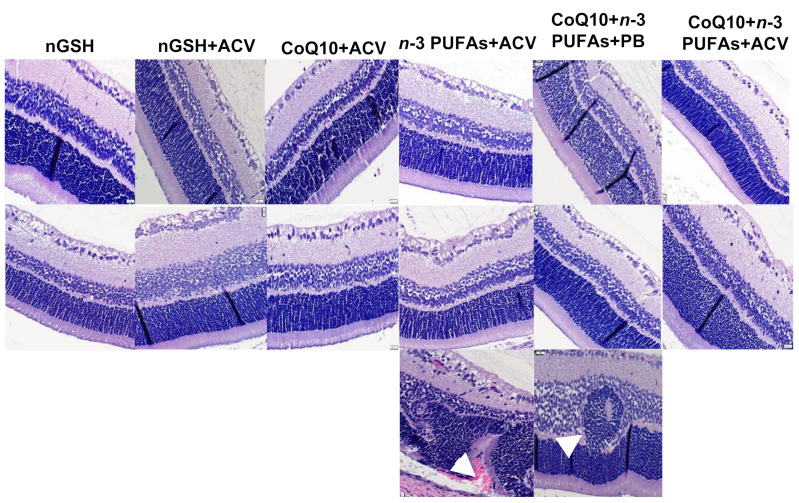
Representative H&E stained retinal layers from neonatal rats on postnatal day 21 (P21). Room air (RA) groups are represented in the top panels and intermittent hypoxia (IH) groups are represented in the bottom panels. Images are 40× magnification, scale bar is 20 µM.

**Table 1 nutrients-12-01980-t001:** % Change in Growth from P0 (birth).

	Weight P7	Length P7	Weight P14	Length P14	Weight P21	Length P21
***nGSH + PBS:***						
RA	127.0 ± 8.4	29.0 ± 1.9	296.7 ± 19.2	92.6 ± 1.8	563.6 ± 17.1	114.2 ± 3.8
IH	78.9 ± 5.3 ^##^	27.4 ± 1.3	190.5 ± 9.7 ^##^	68.9 ± 3.4 ^##^	390.8 ± 18.6 ^##^	97.6 ± 1.8 ^##^
***nGSH + ACV:***						
RA	77.9 ± 2.8 **	28.0 ± 0.98	187.7 ± 4.6 **	56.6 ± 1.2 **	387.3 ± 9.7 **	93.0 ± 2.0 **
IH	92.0 ± 4.4 ^#^	26.5 ± 1.0	196.9 ± 10.4	51.7 ± 2.5 ^§§^	390.0 ± 16.8	88.5 ± 2.4
***CoQ10 + ACV:***						
RA	89.9 ± 5.8 **	33.3 ± 1.4	243.1 ± 9.9 **	61.2 ± 2.1 **	516.3 ± 18.2	104.3 ± 1.5
IH	94.7 ± 4.4 ^#^	27.3 ± 1.9	205.4 ± 6.9 ^##^	49.4 ± 2.6 ^§§^^,^^##^	424.0 ± 15.6 ^##^	87.6 ± 3.5 ^##^
***n-3 PUFAs+ ACV:***						
RA	63.2 ± 4.1 **	24.0 ± 1.1	191.8 ± 6.2 **	54.9 ± 1.1 **	418.1 ± 15.4 **	90.0 ± 2.0 **
IH	73.5 ± 6.4	23.1 ± 1.1	183.7 ± 10.5	45.0 ± 2.0 ^§§^^,^^##^	328.2 ± 17.4 ^##^	83.1 ± 1.8 ^§§^^,^^#^
***CoQ10 + n-3 PUFAs+ PBS:***						
RA	92.2 ± 4.2 **	34.3 ± 1.0 *	222.7 ± 9.3 **	59.2 ± 2.4 **	456.2 ± 21.2 **	93.8 ± 2.7 **
IH	88.9 ± 4.1 ^##^	24.0 ± 1.1 ^##^	213.9 ± 6.5	47.6 ± 1.0 ^§§^^,^^##^	398.9 ± 13.1#	84.9 ± 1.7 ^§§^^,^^##^
***CoQ10 + n-3 PUFAs + ACV:***						
RA	80.2 ± 5.5 **	35.2 ± 1.5 *	209.7 ± 8.8 **	60.4 ± 1.9 **	451.5 ± 17.9 **	97.7 ± 2.2 **
IH	83.3 ± 4.0	22.7 ± 0.94 ^##^	219.7 ± 8.6	56.9 ± 2.3 ^§§^	438.7 ± 16.1	95.7 ± 2.9

nGSH (glutathione nanoparticles); PBS (phosphate buffered saline); ACV (Acuvail); CoQ10 (coenzyme Q10); *n*-3 PUFAs (omega 3 polyunsaturated fatty acids); RA (room air); IH (intermittent hypoxia); P0 (postnatal day 0); P7 (postnatal day 7); P14 (postnatal day 14); P21 (postnatal day 21). Data are mean ± SEM. * *p* < 0.05; ** *p* < 0.01 vs. GSH + PBS-RA; ^§§^
*p* < 0.01 vs. GSH + PBS-IH; ^#^
*p* < 0.05, ^##^
*p* < 0.01 = RA vs. IH. *n* = 18 rats/group.

**Table 2 nutrients-12-01980-t002:** Eye Opening at P14.

	RA			IH		
	Left Eye	Right Eye	Both Eyes	Left Eye	Right Eye	Both Eyes
**nGSH + PBS**	17 (94%)	17 (94%)	17 (94%)	7 (39%) **	8 (44%) **	6 (33%) **
**nGSH + ACV**	18 (100%)	18 (100%)	18 (100%)	12 (67%) **	12 (67%) **	11 (61%) **
**CoQ10 + ACV**	15 (83%)	15 (83%)	14 (78%)	15 (83%)	13 (72%)	13 (72%)
***n*-3 PUFAs + ACV**	14 (78%)	14 (78%)	14 (78%)	8 (47%)	8 (47%)	8 (47%)
**CoQ10 + *n*-3 PUFAs + PBS**	18 (100%)	17 (94%)	17 (94%)	10 (56%) **	8 (44%) **	8 (44%) **
**CoQ10 + *n*-3 PUFAs + ACV**	17 (94%)	17 (94%)	17 (94%)	18 (100%)	16 (89%)	16 (89%)

All animals were examined at P14 (*n* = 18 rats/group; ** *p* < 0.01 vs. RA). On P13, 16/18 rats in the nGSH + ACV group opened both eyes. P14 (postnatal day 14); RA (room air); IH (intermittent hypoxia); nGSH (glutathione nanoparticles); PBS (phosphate buffered saline); ACV (Acuvail); CoQ10 (coenzyme Q10); *n*-3 PUFAs (omega 3 polyunsaturated fatty acids).

**Table 3 nutrients-12-01980-t003:** Retinal Morphometry.

Group	Tortuosity Index	Artery Diameter (µm)	Vein Diameter (µm)	No. Cells in NFL/GCL	Total Retinal Thickness (µm)	NFL/GCL Thickness (µm)	IPL Thickness (µm)	INL Thickness (µm)	ONL Thickness (µm)
***1:***									
**RA**	1.01 ± 0.002	27.6 ± 0.69	40.6 ± 1.5	129.3 ± 13.4	289.5 ± 7.2	33.5 ± 1.4	54.0 ± 1.8	54.0 ± 1.5	91.5 ± 3.4
**IH**	1.03 ± 0.003 ^##^	29.5 ± 0.57 ^#^	41.6 ± 0.95	273.7 ± 20.0 ^##^	290.9 ± 8.5	48.8 ± 2.3 ^##^	59.6 ± 2.1 ^#^	57.8 ± 2.1	83.7 ± 2.9
***2:***									
**RA**	1.02 ± 0.003	28.4 ± 0.63	41.9 ± 1.08	194.4 ± 14.0 **	308.9 ± 8.3	41.2 ± 2.1	53.4 ± 2.0	61.4 ± 2.7	91.5 ± 2.3
**IH**	1.03 ± 0.003 ^#^	28.7 ± 0.61	39.6 ± 1.1 ^##^	304.3 ± 14.8 ^##^	319.5 ± 11.3	51.9 ± 2.2 ^##^	71.8 ± 3.8 ^§^^,^^##^	83.6 ± 3.8 ^§§^^,^^##^	112.1 ± 4.6 ^§§^^,^^##^
***3:***									
**RA**	1.02 ± 0.004	32.9 ± 0.88 **	45.3 ± 1.2	219.3 ± 7.8 **	313.1 ± 12.1	40.8 ± 1.7	54.7 ± 2.5	63.3 ± 3.5	93.5 ± 2.5
**IH**	1.05 ± 0.014 ^##^	28.9 ± 0.62 ^#^	42.8 ± 1.1	296.8 ± 11.0 ^##^	319.9 ± 11.4	52.8 ± 3.2 ^##^	59.6 ± 3.4	66.1 ± 3.3	95.0 ± 2.9
***4:***									
**RA**	1.03 ± 0.006 *	27.8 ± 0.59	47.9 ± 1.3 **	213.6 ± 8.1 **	255.3 ± 5.5	37.6 ± 1.7	48.9 ± 1.8	45.9 ± 1.8	77.9 ± 1.6 *
**IH**	1.07 ± 0.015 ^§§^^,^^##^	29.1 ± 0.63 ^§§^	38.2 ± 0.78 ^##^	275.7 ± 10.8 ^##^	301.7 ± 8.3 ^##^	49.9 ± 1.7 ^##^	56.1 ± 1.9 ^##^	57.9 ± 2.6 ^##^	89.5 ± 2.7 ^##^
***5:***									
**RA**	1.03 ± 0.003 *	26.6 ± 0.63	43.8 ± 1.2	223.6 ± 16.9 **	318.1 ± 8.9	46.3 ± 3.2 **	62.2 ± 3.6	62.0 ± 3.3	93.9 ± 2.9
**IH**	1.08 ± 0.008 ^§§^^,^^##^	29.2 ± 0.65 ^##^	44.2 ± 1.03	265.2 ± 8.4 ^§§^^,^^##^	272.2 ± 5.8 ^##^	45.6 ± 1.5	47.0 ± 1.4 ^##^	51.6 ± 1.4 ^##^	97.7 ± 3.6 ^§^
***6:***									
**RA**	1.02 ± 0.004	29.4 ± 0.64	44.4 ± 1.2	187.9 ± 13.8 **	313.0 ± 13.2	47.9 ± 2.8 **	57.5 ± 1.9	64.9 ± 3.3	89.6 ± 4.4
**IH**	1.03 ± 0.004 ^##^	27.7 ± 0.71	43.8 ± 1.4	204.8 ± 13.3 ^§§^^,^^##^	309.4 ± 5.9	40.3 ± 1.9	59.7 ± 1.8	57.3 ± 1.5#	83.4 ± 2.2

Group 1: nGSH + PBS; Group 2: nGSH + ACV; Group 3: CoQ10 + ACV; Group 4: *n*-3 PUFAs + ACV; Group 5: CoQ10+ *n*-3 PUFAs + PBS; Group 6: CoQ10+ *n*-3 PUFAs + ACV. nGSH (glutathione nanoparticles); PBS (phosphate buffered saline); ACV (Acuvail); CoQ10 (coenzyme Q10); *n*-3 PUFAs (omega 3 polyunsaturated fatty acids); RA (room air); IH (intermittent hypoxia); NFL/GCL (nerve fiber layer/ganglion cell layer); IPL (inner plexiform layer); INL (inner nuclear layer); ONL (outer nuclear layer). Data are mean ± SD; * *p* < 0.05, ** *p* < 0.01 vs. nGSH + PBS RA; ^§^
*p* < 0.05, ^§§^
*p* < 0.01 vs. nGSH + PBS IH; ^#^
*p* < 0.05, ^##^
*p* < 0.01 (RA vs. IH; *n* = 24 measurements/group).

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
