# Peer review of "Combination Antioxidant/NSAID Therapies and Oral/Topical Ocular Delivery Modes for Prevention of Oxygen-Induced Retinopathy in a Rat Model"

_nutrients, 2020, doi:10.3390/nu12071980_

Round 1
Reviewer 1 Report
The author Beharry et al, investigate the effects of a combination of antioxidant/NSAID therapies in different combinations and mode of administration to preventing Oxygen-induced retinopathy in a classical OIR rat model. The author analysed the effects of the different treatment on retinal/choroidal vascular pathology using angiogenesis biomarkers and histopathology experiments and conclude that interesting beneficial effects were observed in OIR rat treated in combination with the different agents. All together, this study is very well constructed with clear results and using appropriate methodology. However, some minor comments are suggested to improve the quality of this study.
Minor comments:
1-General comment; why did you choose the OIR model with a 30min cycling (50% O2) followed by three brief, 1-minute, clustered hypoxic events (12% O2), with a 10-minute re-oxygenation instead of the classic ROP model ( rat OIR cycling; 50% O2 / 10% O2 alternate each 24h for 14days. This model is known to be the rat ROP model closest to humans? Please comments.
2-why in method section 2.4, the eyes were pooled instead of keeping the retina, choroid and vitreous individual of each animal, which would have increased the statistical power with a larger N ?
3-Figure 1; indicated the conditions directly at the top of the western blot, which facilitates the reading of the band. Same think for Figures 6, 7 and 8, replace the letters by the name of the conditions directly on the pictures and add the scale bar on the pictures.
4-Figure 1, how did you explain that the same agent stimulates or inhibits the expression of HIF1a in the retina and has the opposite effect in the choroid? The same phenomenon is observed for the other angiogenic factors detected in the other figures? In the same order of idea, how explained that that certain agent (eg. GSH) decreases the expression of HIF1a in the retina and in the same time increasing the expression of VEGF (or in the same tissue, same phenomenon is observed also the choroid)? HIF1a is VEGF transcription factors?
5- A western blot detecting the activation of VEGFR2 (active form phosphorylated) in the retina and choroid must be added to FIG. 2.
6-An analysis of the impact of the different agents on the thickness of the choroid which has recently been shown to be degenerate in OIR rat (choroidal involution) could be added by staining the vessels with lectin on retinal cross section. Quantification of the choroidal thickness between the different groups are also required.
7- Quantification graph of the % of retinal vasoobliteration and the % of neovascularization on the retinal flatmount (fig.7-8) need to be added.
Author Response
REVIEWER #1:
- We thank the reviewer for his comment regarding the IH model. There are three main reasons for why we use this model in our laboratory. First, the 50-10% O2 model is not clinically relevant. No human neonate (or any human for that matter) will experience 24 hours of 10% O2. Extremely low gestational age neonates experience who are at the highest risk for severe retinopathy of prematurity experience only brief hypoxic episodes (or apneas) during oxygen therapy, lasting less than one minute (Di Fiore JM et al. J Pediatr 2010;157(1):69-73; Martin RJ et al, Neonatology 2011;100(3):303-310). Therefore, our model is more clinically relevant. Second, we conducted an experiment examining the severity of oxygen-induced retinopathy using two paradigms: 1) evenly dispersed brief episodes; and 2) clustered episodes. The results of our experiments showed that the clustered model resulted in a more severe form of OIR (Coleman RJ et al. Pediatr Res 2008;64(1):50-5). This finding was later confirmed in human neonates (Di Fiore JM et al. Pediatr Res 201272(6):606-612). Third, in the neonatal intensive care unit (NICU), the nurses cluster their care in order to avoid frequent handling. During the clustered procedures, there is an increased number of arterial oxygen desaturations. This has been added to the discussion.
- Regarding pooling of the eyes within the groups, this was done to maximize the assay efficiency. The volume of vitreous fluid and the amount of retina and choroidal tissue for each eye is insufficient to run the assays. Each assay requires at least 50 µL sample. The volume of vitreous fluid of both eyes is approximately 20 µL. Similar comment for the retina and choroidal tissues. While we do agree that individual samples would increase the statistical power, the small volume and tissue mass would result in low to undetectable levels. We have tested this and the best strategy is to pool the samples within the same group.
- The Figures have been revised to include the groups instead of letters. The scale bars were burned into all images. However, due to re-orientating the images for clarity, and cropping, the scale bars were deleted in some images but are visible in others. The scale bar is stated in the figure legends.
- There are many reasons for the differences between the responses of the retina and choroid (Figure 1):
- The choroid is primarily a vascular structure that supplies oxygen and nutrients to the outer retina.
- Unlike the inner retina, the choroid is fenestrated (leaky) while the retina has tight junctions. These fenestrae allow easy movement of macromolecules
- The choroid is highly vascularized has a relatively high blood flow, as great as any other organ and blood flow is not autoregulated unlike the retina. Almost the entire blood supply of the eye comes from the choroidal vessels.
- Compared to the retina, the choroid relies on high levels of VEGF (and HIF1α) in order to maintain the fenestrations (VEGF is a vascular permeability factor). Loss of VEGF can result in cone damage.
- Because the choriocapillaris are fenestrated, there is rapid equilibration of drugs present in the blood stream with the extravascular space of the choroid. But the blood-retina barrier and retinal tight junctions restrict the entry from the choroid to the retina.
- The cornea is continuous with the sclera and permeability through the sclera is compatible to that of the cornea (Gaudana R et al. AAPS J 2010;12:348-360). Therefore, topical ocular drug delivery with eye drops (as in the case of Acuvail), may result in drug absorption through the conjunctiva, sclera and choroid (del Amo EL et al. Prog Ret Eye Res 2017; 57:137-185).
Therefore, these differences observed may be due to a number of these factors. This has been clarified in the discussion on page 21, lines 742-748.
- We did not determine VEGFR-2 expression in the retina and choroid by Western blots because it was beyond the scope of our objectives. We determined the soluble splice variants because they are endogenous inhibitors of VEGF action. We do not feel that additional figures of VEGFR-2 Western blots will change the outcomes or conclusions.
- We agree with the reviewer that choroidal involution is important in OIR and is found to be associated with myopia in later life. However, the focus of this work is the retina and whether non-invasive therapies are safe and effective for preventing retinal angiogenesis. Furthermore, these experiments were completed several months ago. Lectin staining would require fresh retinas. In keeping with the regulations of our Institutional Animal Care and Use Committee, we cannot increase animal numbers for this purpose.
- Unlike OIR in mice, rats (like human preterm infants), do not have vasoobliteration of retinal central vessels (see Grossniklaus HE et al. Prog Retin Eye Res 2010;29(6):500-519). However, vessel density was calculated at the periphery as a measure of vasopriliferation during the recovery-reoxygenation phase at P21. The methods are reported on page 7 lines 378 & 279, and the results on page 16, lines 531-535.
Reviewer 2 Report
In this manuscript Beharry et al investigated the potential of a combination of antioxidants and oral/topical delivery systems in the oxygen induced retinopathy models. The article is well written images are of good quality. Here are some of my concerns regarding this article.
Major Comments
- In the experimental design section; line 137, the authors mentioned about the IH induction process. Is this being the globally accepted model, if yes cite the reference. If this technique is developed by the authors, mention that as well.
- How did you fix the dose of all the nutrients?
- IH induced from P0 to P14, did that was fatal to the pups, indicate the mortality rate
- In the line 583, it is written that image acquisition done at 20X magnification for figures 9 and 10 and it is written as 10X in the foot note of those figures. Scale bars are also different.
- In the figures 9 and 10, indicate which color is GFAP and isolectin B4 and also about the merge color.
Minor Comments
- In the line 110, there should be the expanded form of NSAID in brackets.
- Why the subsection 3.2 and 3.5 is given to Table 1 and 2 respectively (Line number 323, 360) and such a classification was not seen on Table 3 (Line number 573)
Author Response
REVIEWER #2:
- We thank the reviewer for his/her comments regarding our neonatal IH model. As stated above, this model is more clinically relevant than most of the models used for studying OIR. The 50-10% model uses 24 hours of 50% O2 followed by 24 hours of 10% O2. This is not consistent with brief, alternating hypoxia (apnea) during oxygen therapy experienced by preterm infants at risk for severe ROP. Our model is an improvement of the 50-10% model and was developed by our group. We have many publications using this model. This point was mentioned on page 5 line 173.
- The doses of all the Acuvail, glutathione nanoparticles, were based on the manufacturer’s recommendations and adjusted for the neonatal rat. The doses of coenzyme Q10, and n-3 PUFAs was based on our previous findings (reference #8). See page 4.
- The reviewer raises a great question regarding the mortality rate. In this model, the survival rate is 100%. Due to the brief hypoxic episodes (1-minute) with 10 minutes of hyperoxia between each episode, this does not result in any mortality. Furthermore, each day, the animals are taken out for a period of about 5 minutes for supplementation and injection; and the dams are switched between room air and IH every 3 days.
- We thank the reviewer for pointing out the discrepancy on line 583. The legend of the images and scale bar are corrected.
- The color of GFAP (green) and isolectin B4 (red) has been identified in the text and in the figure legends.
- We have spelled out NSAIDs.
- We have corrected the subsection for Table 3 to be consistent with Tables 1 and 2.